# Artificial SiN_z_:H Synapse Crossbar Arrays with Gradual Conductive Pathway for High-Accuracy Neuromorphic Computing

**DOI:** 10.3390/nano13162362

**Published:** 2023-08-18

**Authors:** Tong Chen, Zhongyuan Ma, Hongsheng Hu, Yang Yang, Chengfeng Zhou, Furao Shen, Haitao Xu, Jun Xu, Ling Xu, Wei Li, Kunji Chen

**Affiliations:** 1School of Electronic Science and Engineering, Nanjing University, Nanjing 210093, China; 2Collaborative Innovation Center of Advanced Microstructures, Nanjing University, Nanjing 210093, China; 3Jiangsu Provincial Key Laboratory of Photonic and Electronic Materials Sciences and Technology, Nanjing University, Nanjing 210093, China; 4School of Artificial Intelligence, Nanjing University, Nanjing 210093, China; 5Institute of Advanced Functional Materials and Devices, Shanxi University, Taiyuan 030031, China; 6Institute of Carbon-Based Thin Film Electronics, Peking University, Taiyuan 030031, China

**Keywords:** artificial synapse, multilevel resistive switching, silicon nitride, Si dangling bond

## Abstract

Inspired by its highly efficient capability to deal with big data, the brain-like computational system has attracted a great amount of attention for its ability to outperform the von Neumann computation paradigm. As the core of the neuromorphic computing chip, an artificial synapse based on the memristor, with a high accuracy in processing images, is highly desired. We report, for the first time, that artificial synapse arrays with a high accuracy in image recognition can be obtained through the fabrication of a SiN_z_:H memristor with a gradient Si/N ratio. The training accuracy of SiN_z_:H synapse arrays for image learning can reach 93.65%. The temperature-dependent *I*–*V* characteristic reveals that the gradual Si dangling bond pathway makes the main contribution towards improving the linearity of the tunable conductance. The thinner diameter and fixed disconnection point in the gradual pathway are of benefit in enhancing the accuracy of visual identification. The artificial SiN_z_:H synapse arrays display stable and uniform biological functions, such as the short-term biosynaptic functions, including spike-duration-dependent plasticity, spike-number-dependent plasticity, and paired-pulse facilitation, as well as the long-term ones, such as long-term potentiation, long-term depression, and spike-time-dependent plasticity. The highly efficient visual learning capability of the artificial SiN_z_:H synapse with a gradual conductive pathway for neuromorphic systems hold great application potential in the age of artificial intelligence (AI).

## 1. Introduction

With the artificial intelligence era of ChatGPT approaching, the development of a next-generation brain-like computing chip with high efficiency and accuracy is urgently needed for dealing with massive and diverse data. It requires the combination of memory and computing functions integrated in a unit [1], which is different from the traditional computer system with its physically separated memory module and computing module [2,3,4]. As the basic elements of the brain’s computing system, synapses are in charge of performing memory- and learning-related tasks through synaptic plasticity, which depends on the tunable synaptic weight. The variation of synaptic weight under stimuli is similar to the conductance of a memristor under an electric field. Thus, constructing artificial synapses based on a memristor with a controlled multilevel conductance to simulate efficient biological information processing is vitally significant [5,6,7]. Although progress in the optimization of electronic artificial synapses has been made in oxide-based memristors [5,6,7,8,9,10,11,12,13,14,15,16,17,18], the major challenge is still to design Si-based artificial synapses with high accuracy due to their perfect compatibility with CMOS technology. As for the memristors with a conductive pathway, the resistive behavior can be attributed to the random breakage of the filament under the action of an electric field. To mimic the synaptic characteristic with high accuracy, the gradual change in resistance under high-resistance and low-resistance states is vitally important. Up until now, there has been no report on Si-based memristors with a gradual conductive filament path formed in a resistive switching layer with a gradient chemical composition.

In this work, we fabricated memristive crossbar arrays based on silicon nitride (SiN_z_:H) films with a gradual ratio of N/Si by changing the SiH_4_/NH_3_ gas ratio in the PECVD system. The N/Si ratio of SiN_z_:H films varies from 0.69 to 1.41 upwards from the bottom to the surface of the SiN_z_:H memristor. By tuning the reset voltage, the SiN_z_:H memristive crossbar array exhibits five stable resistance states with stable endurance and retention properties. The theoretical fitting analysis indicates that the conduction behaviors in five resistance states follow the Schottky conduction mechanism. In addition, the Schottky barrier increases from 0.40 V to 0.84 V with the reset voltage increasing. Based on the analysis of the resistance–temperature dependence, it is revealed that the evolution of the Si dangling bond (DB) conduction pathway under an electric field is the origin of the multilevel RS characteristics of the SiN_z_:H films. It is noteworthy that biosynaptic functions including long-term potentiation (LTP), long-term depression (LTD), spike-time-dependent plasticity (STDP), and paired-pulse facilitation (PPF) are implemented successfully in the SiN_z_:H-based device. Moreover, the feasibility of our artificial synapse for neuromorphic systems is demonstrated, which shows its great application potential in synaptic neural networks with a highly efficient learning capability. A gradient conductive pathway means continuous changes in concentration for the filament, which supplies an excellent platform to realize the gradually changed electrical properties. The training accuracy of SiN_z_:H synapse arrays for image learning can reach 93.65%. The high learning accuracy of artificial synapses with a gradual filament supplies a new avenue for neuromorphic computing in the age of artificial intelligence (AI).

## 2. Materials and Methods

Firstly, SiO_2_ films with a thickness of 300 nm were grown on the top of Si wafers using the wet thermal oxidation method. Secondly, the surface of the SiO_2_/p^+^-Si substrates was patterned by standard photolithography. Then, the bottom electrodes (BEs) were formed by thermal evaporation of Al on the patterned area of SiO_2_, followed by lift-off processes. Thirdly, the SiN_z_:H films were deposited on the surface of the BE by decomposing SiH_4_ and NH_3_ in a PECVD system. The SiN_z_:H films with gradual Si/N ratio can be obtained by changing the gas flow ratio of SiH_4_ to NH_3_ from 8/20 to 8/40 gradually during the deposition process. Fourthly, the SiN_z_:H films deposited on the pad of Al BEs were removed by using the GSE etching system. Finally, the top electrodes (TEs) with diameters of 200 μm were obtained by thermal evaporation of Al to the SiN_z_:H films using the standard photolithography and lift-off processes. For comparison, the SiN_x_:H films with a fixed ratio of Si/N (reference sample) were fabricated by decomposing SiH_4_ and NH_3_ with a fixed gas flow ratio of 8/40. The microstructure of the Al/SiN_z_:H/p^+^-Si device was investigated by high-resolution transmission electron microscopy (HRTEM, JEOL 2100F). The atomic bonding configuration of SiN_z_:H films was analyzed through FTIR spectroscopy (NEXUS870). To determine the N/Si atomic concentration ratios of the SiN_z_ films, the XPS measurements were carried out using a PHI 5000 VersaProbe. The electrical measurements of the devices were performed by an Agilent B1500A semiconductor analyzer. The temperature-dependent *I*–*V* characteristic measurement was carried out by using the Lake Shore CRX-4K system.

## 3. Results and Discussion

A microstructure diagram of the Al/SiN_z_:H/Al/SiO_2_/p^+^-Si resistive switching crossbar arrays is depicted in Figure 1a. The SiN_z_:H films are sandwiched between the Al TEs and Al BEs. Figure 1b shows the optical microscope image of the SiN_z_:H-based resistive switching crossbar arrays. The crosspoint image is amplified, as shown in Figure 1c. A diagram of SiN_z_:H films with a gradient atomic N/Si ratio is presented in Figure 1d, which decreases from 1.14 to 0.36 with the etched depth of the SiN_z_:H films increasing from the surface to the bottom. The HRTEM image of the Al/SiN_z_:H/Al cross-point is shown in Figure 1e. The thickness of the SiN_z_:H layer is 8 nm. The two interfaces between the SiN_z_:H layer and the two electrodes are abrupt. The FTIR spectra of the as-deposited SiN_z_:H films with a gradient N/Si atomic ratio from 0.69 to 1.41 and the SiN_x_:H films with an atomic ratio N/Si of 1.41 are shown in Figure 1f. The Si-N stretching mode, N-H wagging mode, Si-H stretching mode, and N-H stretching mode can be found in the absorption bands of the two kinds of films, which are located at 841 cm^−1^, 1176 cm^−1^, 2173 cm^−1^ and 3356 cm^−1^, respectively [19,20]. Si-H stretching and N-H stretching modes are bound up with the hydrogenation of silicon nitride.

In order to explore the element distribution of SiN_z_:H films with a gradual ratio of Si/N at different depths, the following four conditions of Ar etching were used to treat SiN_z_:H films: (1) 1 kV, 30 s; (2) 1 kV, 40 s; (3) 2 kV, 30 s; (4) 2 kV, 40 s. Figure 1g shows the Si 2*p* and N 1*s* XPS spectra of the SiN_z_:H films at five different etching depths. According to the XPS depth profiles, the ratios of N/Si at five different depths are 1.41, 1.1, 0.96, 0.82 and 0.69, respectively. The atomic ratio of N to Si at five different etching depths is shown in Figure 1h. It is evident that the atomic percentage of Si element increases with the increasing etching depth. and the atomic percentage of N decreases with the decreasing etching depth. It is further confirmed that the N/Si ratio of the SiN_z_:H films is gradient. Figure 2a displays the electric measurement diagram of the Al/SiN_z_:H/Al/SiO_2_/p^+^-Si memristor crossbar arrays. The bias was applied to the Al TEs with the BEs grounded. To protect the SiN_z_:H memristor from breakdown, a compliance current of 1 mA was used during the *I*–*V* measurement. When testing the electrical characteristics of a single device, the unselected devices in the 5 × 5 crossbar array can be kept in the high-resistance state by applying a lower voltage so as to avoid cross-talk interference. Figure 2b shows the *I*–*V* curves of the SiN_z_:H memristor after 100 consecutive cycles. A typical bipolar RS behavior is observed with a forming voltage of 7 V. When a positive bias was applied to the Al TE, a sharp set process occurred in the device. When a negative bias voltage was applied to the Al TE, the device displayed a gradual reset process. Compared with the forming voltage, the subsequent set voltage was reduced to 5.3 V. The reset voltage was approximately −3.75 V. The powers for SET and RESET operations were 0.0150 W/cm^2^ and 0.0079 W/cm^2^, respectively. It is worth noting that the *I*–*V* curves of the SiN_z_:H memristor unit during the following 100 cycles overlapped with each other, displaying its stable resistive switching characteristic. Figure 2c shows the *I*–*V* curves of the SiN_x_:H-based memory device under the initial 100 consecutive cycles. It is worth noting that the SiN_x_:H-based memory device has a sharp reset process, which is completely different from that of the SiN_z_:H-based devices. In the above-mentioned electrical measurements, we found that a gradual reset process occurs in the Al/SiN_z_:H/Al/SiO_2_/p^+^-Si device, indicating its potential in a multilevel storage. Next, the multilevel RS memory performance was evaluated. The *I*–*V* characteristics of the SiN_z_:H device under different RESET voltages were investigated, which were compared with those of the SiN_x_:H device.

As shown in Figure 2d, the current of the SiN_z_:H device keeps decreasing gradually with the gradual increase in negative RESET voltages. In contrast, a sudden drop in current can be detected from the SiN_x_:H device when the negative RESET voltage reaches a threshold, as shown in Figure 2e. With the increase in negative RESET voltages, the current decreases sharply. To evaluate the multilevel RS memory property of the Al/SiN_z_:H/Al/SiO_2_/p^+^-Si device, the endurance test was carried out, as shown in Figure 2f. To achieve the multilevel RS, the RESET voltage was tuned from −1.32 V to −1.68 V with a constant read voltage of 0.5 V. At the beginning, the set voltage of +4.7 V was applied to convert the device to the LRS. Multilevel RS was obtained by tuning the RESET voltage successively from −1.32 V to −1.48 V, −1.60 V, and −1.68 V. As shown in Figure 2f, a uniform multilevel RS property and an adequate separation among these RS states are achieved for the Al/SiN_z_:H/Al/SiO_2_/p^+^-Si device. To evaluate the stability of all five different resistance states, retention measurements were performed, as shown in Figure 2g. It can be observed that the five memory states are stable and can be retained within 10^4^ s with no degradation.

To investigate the carrier transportation rules of the SiN_z_:H memristor in different resistance states, representative *I*–*V* curves in the positive bias region were fitted using the following Schottky diffusion current equation [21,22,23]:(1)$$J=A^{*}T^{2}\exp(\frac{q\sqrt{qE/4\pi \varepsilon_{i}}-q\Phi_{B}}{k_{B}T}),$$
where *J* is the current intensity, *E* is the electric field intensity, *T* is the temperature (300 K), *q* is the elementary charge, $\varepsilon_{i}$ is the insulation permittivity, $\Phi_{B}$ is the Schottky barrier, $k_{B}$ is the Boltzmann’s constant, and *A** is the Richardson’s constant. As shown in Figure 2h, the experimental data at different resistance states are in good agreement with the fitting results. Moreover, it was found that the Schottky barrier of the SiN_z_:H memristor decreases from 0.45 eV to 0.38 eV with the reset voltage increasing. In order to explore the composition of the conductive pathway in the Al/SiN_z_:H/Al/SiO_2_/p^+^-Si device, we measured the current changes in the HRS and LRS at different temperatures, as shown in Figure 2i. The current intensity of HRS and LRS was enhanced with the temperature increasing, showing semiconductor properties. It rules out the contribution of metal filament to the resistive switching. Considering the semiconductor characteristic of temperature-dependent *I*-*V* curves from the SiN_z_:H memristor, the resistive switching is related with the connection and breakage of the Si dangling bond pathway between the TE and BE. As revealed by the FTIR in Figure 1, there was a number of Si–H bonds in the as-deposited SiN_z_:H films. Compared with that of the Si–N and N–H bonds, the bonding energy of the Si–H bond is the lowest one, which is easier to be broken under the set voltage. The XPS spectra indicate that the number of Si–H bonds decreases gradually from the bottom to the top of the films. Thus, the number of Si dangling bonds increases from TE to BE under the set voltage, forming a gradient Si dangling pathway. When the reset voltage is applied to the device, H^+^ returns to passivate Si DBs under the reverse electric field, disconnecting the Si DB conduction pathway. The initial breakage position of the conductive pathway is near the TE. With the reset voltage increasing, the number of Si DBs in the conductive pathway is decreased due to the passivation of H^+^. The higher the reset voltage, the larger the number of Si–H bonds [24,25]. Therefore, the current induced by the Si dangling bond is decreased with the reset voltage increasing.

Figure 3a displays statistical distributions of SET and RESET currents of 20 different devices. It is obvious that the operating currents of these devices show good uniformity. Figure 3b shows the current values of the SiN_z_:H-based memristor with different device areas in the HRS and LRS. It can be seen that when the device area increases from 100 µm^2^ to 800 µm^2^, the HRS and LRS show small resistance fluctuations, indicating the existence of local conductive filaments. According to the above analyses, a resistive switching pathway model of the Al/SiN_z_:H/Al/SiO_2_/p^+^-Si device is proposed as shown in Figure 3c. When the first positive voltage is applied to the Al electrode (Set 1), a number of Si DBs are produced due to the broken Si–H bonds, which switches the device to the LRS. When a negative bias is applied to the Al electrode (Reset 1), a large number of Si DBs are passivated by H^+^ ions, switching the device to the HRS. When the second positive voltage is applied to the Al electrode (Set 2), the device returns to the LRS, and the Si DB conductive channel is formed again. Subsequently, a smaller negative bias is applied to the Al electrode to complete the Reset 2 operation. Compared with the Reset 1 operation, less Si DBs are passivated by H^+^ ions in the Reset 2 operation. Therefore, the device enters the intermediate-state MRS3. Following the above operation with smaller reset voltages, the device can also be reset to the intermediate resistive states of MRS2 and MRS1. By tuning the reset voltage, the Al/SiN_z_:H/Al/SiO_2_/p^+^-Si device can exist in different intermediate states.

According to the response retention time, the synaptic plasticity can be classified as long-term memory (LTM) and short-term memory (STM). We first studied the STM of the Al/SiN_z_:H/Al synapse, including spike-duration-dependent plasticity (SDDP), spike-number-dependent plasticity (SNDP) and paired-pulse facilitation (PPF). A diagrammatic sketch of the pre-synapse and the post-synapse based on the Al/SiN_z_:H/Al memristor is shown in Figure 4a. The representative temporal correlation between continuous spiking pulses is the PPF, which can be implemented through applying two continuous positive pulses (1.2 V, 200 μs). Figure 4b shows the excitatory postsynaptic current (EPSC) acting as the premise for the variation of biosynaptic weight by detecting the timely response to two external pulse signals (1.2 V, 200 μs). Here, a read voltage of 0.1 V was applied repeatedly on the memristive device to trace the relaxation of the conduction. The current of the memristive device can remain a higher value for 290 μs even after the pulse signal was withdrawn, which is similar to biosynapse. As shown in Figure 4b, the current triggered by the second positive pulse is distinctly larger than that triggered by the first pulse, which is analogous to the characteristics of biosynapses. This function is important for biological synapses in processing information in the frequency domain. As displayed in Figure 4c, we further studied the temporal correlation of PPF by collecting the peaks of both currents. The pulse waveform for measuring the PPF in the Al/SiN_z_:H/Al synapse is shown in the inset of Figure 4c. The PPF ratio can be obtained by the following Equation (2):(2)$$\mathrm{PPF}\frac{G_{2}-G_{1}}{G_{1}}\times 100\%,$$
where *G*_1_ and *G*_2_ refer to the device conductance after the first spike and the second spike, respectively [26]. When the time interval of pulse training decreases, the potentiating effect of the pre-synaptic spike on the subsequent spike is enhanced, which is similar to the behavior of a biosynapse. The SDDP of the Al/SiN_z_:H/Al synapse is shown in Figure 4d. The pre-synaptic spike duration varies from 20 to 65 μs (Δt = 5 μs) and the amplitude of the spike is fixed at 1.20 V. A_1_ represents the current after the first pulse. The weight variation of the SiN_z_:H-based synapse can be characterized by the SDDP index (A_n_/A_1_ × 100%; *n* = 1, 2, 3…, 10), as shown in Figure 4e. It is shown that a transitory enhancement of the biosynaptic weight was produced by a progressive spike duration. To avoid the influence of previous pulses, the devices were reset to the same initial state before the application of a new pulse, which is similar to the change in the biosynaptic weight. As shown in Figure 4f, we further explored the SNDP of the Al/SiN_z_:H/Al/SiO_2_/p^+^-Si device. The number of the pre-synaptic spike varies from 1 to 10 and the amplitude of the spike is fixed at 1.20 V. The time intervals between the spikes and the spike duration are 500 μs and 20 μs, respectively. A_1_ represents the current after the first pre-synaptic spike. The weight variation of the SiN_z_-based synapse under the incremental spikes is confirmed by the SNDP index (A_n_/A_1_ × 100%; *n* = 1, 2, 3…, 10), as shown in Figure 4g. A transient enhancement of the synaptic weight is produced as the spike number is increased gradually from 1 to 10.

Different from the STM, a permanent synaptic response is formed in the LTM, which consists of long-term potentiation (LTP) and long-term depression (LTD). The LTM features of the SiN_z_:H-based synapse were verified by applying 25 positive pulses followed by 25 negative pulses, as shown in Figure 4h. The amplitude and duration of each pulse remained at 1.2 V and 200 μs, respectively. It is shown that the conductance of the SiN_z_-based synapse increased/decreased when continuous positive/negative spikes were applied. The permanent updating of the conductance was tested under the triggering of 250 consecutive pulses, exhibiting a highly uniform analogue switching. Spike-timing-dependent plasticity (STDP) is also an essential and typical kind of the LTM, which indicates that the connection strength of the neurons can be controlled by tuning the relative timing between pre-spiking and post-spiking [27]. Herein, the STDPs were detected in our Al/SiN_z_:H/Al/SiO_2_/P^+^-Si device by applying the pre-synaptic and post-synaptic input spikes for different temporal intervals, ranging from 0 to 600 µs. A biomorphic-action-potential-like waveform is used to implement the STDP function, as shown in the inset of Figure 4i. When the pre-synaptic input spike stimulates earlier than the post-synaptic input spike (Δt > 0), the connection strength of the Al/SiN_z_:H/Al synapse increases, displaying the characteristic of the LTP [28]. When the post-synaptic input spike stimulates earlier than the pre-synaptic input spike (Δt < 0), the connection strength of the Al/SiN_z_:H/Al synapse decreases, presenting the characteristic of the LTD [29]. The STDP reflects the connection strength of the Al/SiN_z_:H/Al synapse between the two neurons by modulating the relative timing (Δt) of activations from the pre neuron to the post neuron, which is in accordance with the Hebbian learning rule.

In order to facilitate the application of the synaptic device in image learning, the training process of the artificial synapse based on the Al/SiN_z_:H/Al/SiO_2_ synapse arrays was studied by employing the LTP and LTD characteristics under the pulse amplitude of 4.0 V. Figure 5a shows the schematic illustration of the 6 × 6 artificial synapse crossbar arrays for pattern recognition, in which the Al TE acts as the pre-synapse and the Al BE acts as the post-synapse. Three target visual patterns used for image memorization are displayed in Figure 5b. A learning process of the three target visual patterns is shown in Figure 5c. The conductance values represented by the color gradation ranges from 1.3 × 10^−7^ to 2.3 × 10^−5^ S. Initially, all the artificial synapses are triggered to a random conductance state, which corresponds to different colors. The consecutive stimulus pulses were input to the synapse arrays in correspondence of the predefined pattern. Patterns 1, 2 and 3 are submitted in our neuromorphic array to test the learning ability of memorizing different images. A good accordance of the synapse network to memorize Patterns 1, 2, and 3 is demonstrated by the weight maps after 30 epochs, respectively.

Figure 5d,f,h show the normalized conductance distribution and nonlinearity fitting curves of the Al/SiN_z_:H/Al synapse consisting of 6 × 6 units under the pulse amplitude of 4.0 V, 3.0 V and 2.0 V, respectively. The pulse width remains 200 μs. The nonlinearity of the Al/SiN_z_:H/Al/SiO_2_/P^+^-Si synapse decreases from 0.94 to 0.73 with the pulse amplitude changing from 4.0 to 2.0 V. To calculate the nonlinearity (NL) value of the weight update, the following equation was employed:*NL* = *Max*|*GP*(*n*) − *GD*(*n*)|for *n* = 0.1 to 1, (3)
where *G_P_* represents the conductance values after applying the *n*th potentiation pulse and *G_D_* represents the conductance values after applying the *n*th depression pulse. These values are normalized, ranging from 0 to 1 during the weight update. *NL* is equal to zero when the weight update is completely linear. We define the accuracy of the pattern as follows:(4)$$Accuracy=\frac{G_{P}(n)+(1-G_{D}(n))}{2a}\times 100\%\;\mathrm{for}\;n\;=\;1\;\mathrm{to}\;K,$$
where *G_P_*(*n*) represents the normalized conductivity of synaptic pixels after applying the positive excitation pulses, *G_D_*(*n*) represents the normalized conductivity of synaptic pixels after applying the negative excitation pulses, and *K* represents the number of the pixels. When the pulse amplitude decreases from 4.0 V to 3.0 V and 2.0 V, the training accuracy of the Al/SiN_z_:H/Al/SiO_2_/P^+^-Si synapse arrays for the image learning increases from 86.23% to 93.65%, which is corresponding to the reduced nonlinearity from 0.94 to 0.73. It indicates that the high linearity can significantly improve the fault tolerance rate of neuromorphic calculation. To illustrate the contribution of the gradual Si dangling bond pathway to the linearity, the conductive pathway of the SiN_z_:H synapse with a gradient Si/N ratio and the SiN_x_:H synapse with a constant Si/N ratio is compared, as shown in Figure 5j,k. As for the SiN_x_:H synapse with a constant Si/N ratio, when a positive bias is applied on the top electrode, hydrogen ions can move toward the bottom electrodes to form the conductive pathway of Si DBs. It makes the SiN_x_ memristor switch from the HRS to the LRS. As the formation of the conductive pathway is stochastic in the SiN_x_:H memristor, the second conductive path after the first cycle is different from the first one, which results in the poor uniformity and stability during the resistive switching process, as displayed in Figure 5j. Under the negative bias, the hydrogen ions can migrate back toward the top electrode to passivate the Si dangling bonds, disconnecting the Si dangling bond pathway. It switches the device from the LRS to the HRS. As for the SiN_z_ memristor with a gradient Si/N ratio, a gradual Si dangling bond pathway can be formed in the SiN_z_:H layer when the positive bias is applied on the top electrode. Owing to the Si/N ratio decreasing from the top electrode to the bottom electrode, the number of the Si dangling bonds near the top electrode is less than that of the other part of the conductive pathway, as shown in Figure 5k. It makes the diameter of the Si dangling bond pathway near the top electrode the thinnest, which is easier to be broken up under the negative bias because the hydrogen ions could migrate toward the top electrodes to passivate the Si dangling bonds. The fixed position for disconnecting of conductive pathway is beneficial to improving the linearity of the SiN_z_:H synapse, which is the key to the high accuracy of neuromorphic computing. It is worth noting that the computing accuracy of the SiN_z_:H synapse increases from 86.23% to 93.65% with the amplitude of the pulse decreasing from 4.0 to 2.0 V. The reason lies in the formation of the thinner gradual Si dangling bond pathway because the number of Si dangling bonds decreases under the lower voltage. The diameter of the conductive pathway is determined by the number of Si dangling bonds. The thinner the conductive pathway, the lesser the number of Si dangling bonds. The thinner conductive pathway with the fixed disconnection point supplies a good way for the improved linearity. Therefore, the high accuracy of the SiN_z_:H synapse network can be realized.

## 4. Conclusions

In summary, we successfully obtained an artificial SiN_z_:H synapse with a high accuracy in processing images based on the SiN_z_:H films with a gradient Si/N ratio. The training accuracy of SiN_z_:H synapse arrays for image learning can reach 93.65%. The temperature-dependent *I*–*V* characteristic reveals that the gradual Si dangling bond pathway makes the main contribution towards improving the linearity of the tunable conductance. The thinner diameter and fixed disconnection point in the gradual pathway are of benefit in enhancing the accuracy of visual identification. The artificial SiN_z_:H synapse arrays display stable and uniform biological functions, such as the short-term biosynaptic functions, including spike-duration-dependent plasticity, spike-number-dependent plasticity, and paired-pulse facilitation, as well as the long-term ones, such as long-term potentiation, long-term depression, and spike-time-dependent plasticity. The highly efficient learning capability of the artificial SiN_z_:H synapse arrays with a gradual conductive pathway for image recognition holds great application potential in the age of artificial intelligence.

## Figures and Tables

**Figure 1 nanomaterials-13-02362-f001:**
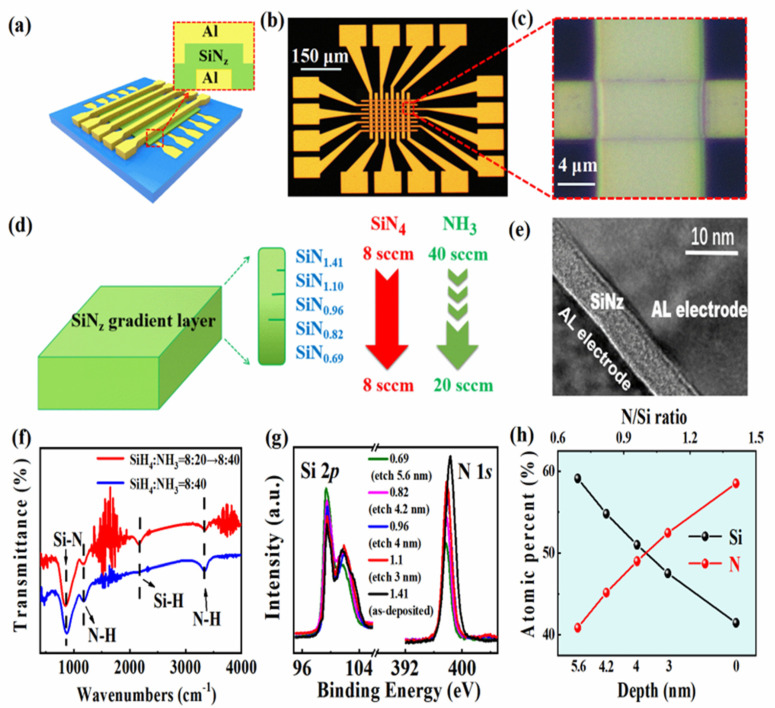
(**a**) A microstructure diagram of SiN_z_:H–based resistive switching crossbar arrays. (**b**) An optical microscope image of the SiN_z_:H-based resistive switching crossbar arrays. (**c**) The crosspoint image of the SiN_z_:H-based memristor crossbar arrays. (**d**) A diagram of the SiN_z_:H films with gradient atomic N/S ratio from 0.69 to 1.41. (**e**) An HRTEM image of the Al/SiN_z_:H/Al crosspoint with gradient atomic N/Si ratio from 0.69 to 1.41. (**f**) FTIR spectra of the as-deposited SiN_z_:H film with gradient N/Si atomic ratio from 0.69 to 1.41 and the SiN_x_:H films with fixed N/S atomic ratio of 1.41. (**g**) The Si 2*p* and N 1*s* XPS spectra of the SiN_z_:H films at five different etching depths. (**h**) The atomic percentage of Si and N in the SiN_z_:H films at five different etching depths.

**Figure 2 nanomaterials-13-02362-f002:**
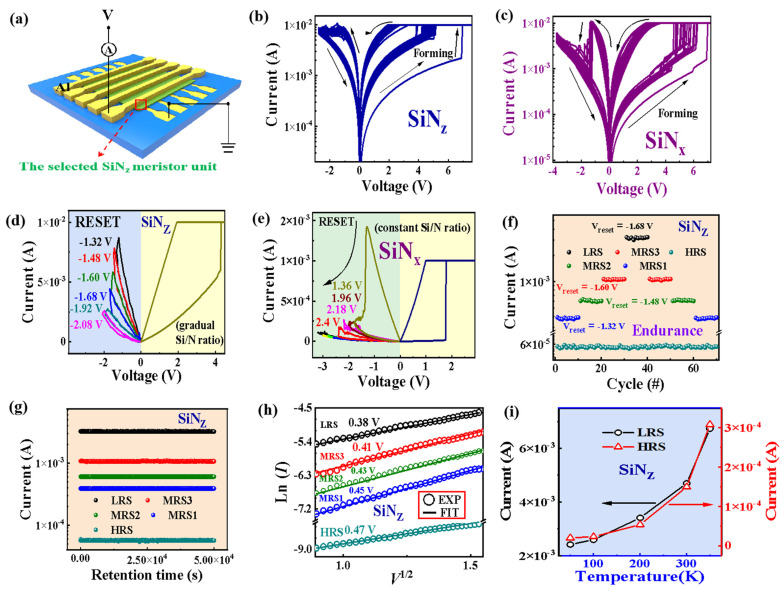
(**a**,**b**) A diagram and *I*–*V* curves of the SiN_z_:H memristor crossbar arrays with gradient N/Si atomic ratio from 0.69 to 1.41. (**c**) *I*–*V* curves of the SiN_x_:H memristor with constant N/Si atomic ratio. (**d**) Multilevel resistive switching characteristics of the SiN_z_:H memristor with different reset voltages. (**e**) Multilevel resistive switching characteristics of the SiN_x_:H memristor with different reset voltages. Yellow, black and green lines represent hysteresis curves that are not very obvious under high RESET voltages. (**f**,**g**) The endurance and retention characteristics corresponding to five selected resistive switching state of the SiN_z_:H memristor by tuning the reset voltage from −1.32 to −1.68 V. (**h**) Experimental and theoretical fitting of *I*–*V* curves corresponding to the five resistive switching states of the SiN_z_:H memristor according to the Schottky model. (**i**) Temperature-dependent current of the SiN_z_:H memristor at high resistance states and low resistance states.

**Figure 3 nanomaterials-13-02362-f003:**
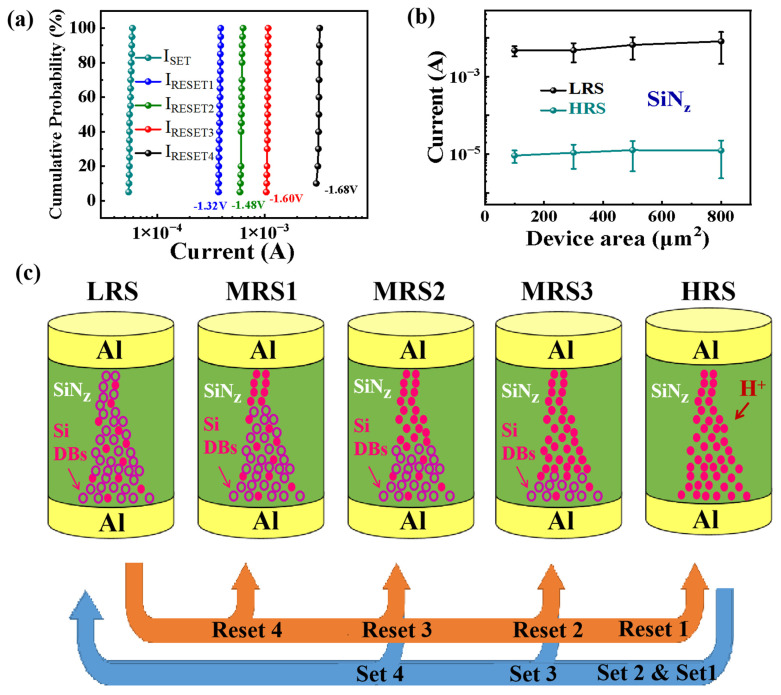
(**a**) Statistical distributions of SET and different RESET currents of 20 SiN_z_ devices. (**b**) The current values of the SiN_z_:H-based memristor with different device areas in the HRS and LRS. (**c**) A schematic of multilevel resistive switching model for the Al/SiN_z_:H/Al/SiO_2_/p^+^-Si device with gradient atomic N/Si ratio from 0.69 to 1.41. The hollow circle represents silicon dangling bonds (Si DBs), and the solid circle represents hydrogen ions.

**Figure 4 nanomaterials-13-02362-f004:**
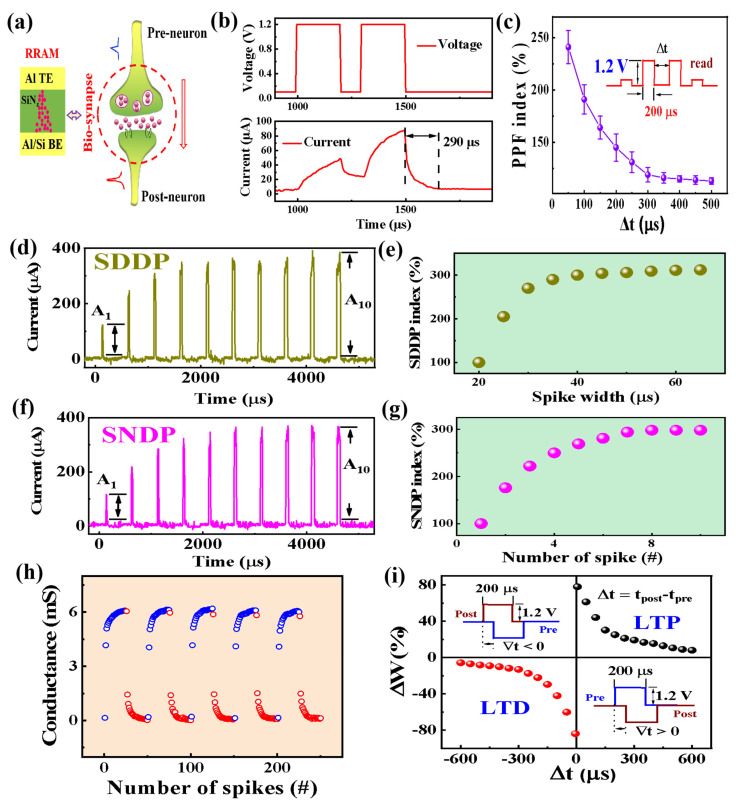
(**a**) A diagrammatic of Al/SiN_z_:H/Al synaptic plasticity between the presynaptic neuron and the postsynaptic neuron. (**b**) The current response triggered through applying two continuous positive pulses (1.2 V, 200 μs). (**c**) The PPF characteristics of the Al/SiN_z_:H/Al device and spikes for PPF measurement. (**d**) The SDDP characteristics of the Al/SiN_z_:H/Al device with the spike duration from 20 to 65 μs. (**e**) The SDDP index (An/A1∙100%; *n* = 1, 2, 3…, 10) of the Al/SiN_z_:H/Al/SiO_2_/P^+^-Si device. (**f**) The SNDP characteristics of the Al/SiN_z_:H/Al device with the spike numbers *n* = 1, 2, 3…, 10. (**g**) The SNDP index (An/A1∙100%; *n* = 1, 2, 3…, 10) of the Al/SiN_z_:H/Al device with the spike numbers *n* = 1, 2, 3…, 10. (**h**) The conductance potentiation and depression through pulse programming. The conductance value was read at 0.1 V after each programming pulse. (**i**) The STDP characteristics of the Al/SiN_z_:H/Al synapse.

**Figure 5 nanomaterials-13-02362-f005:**
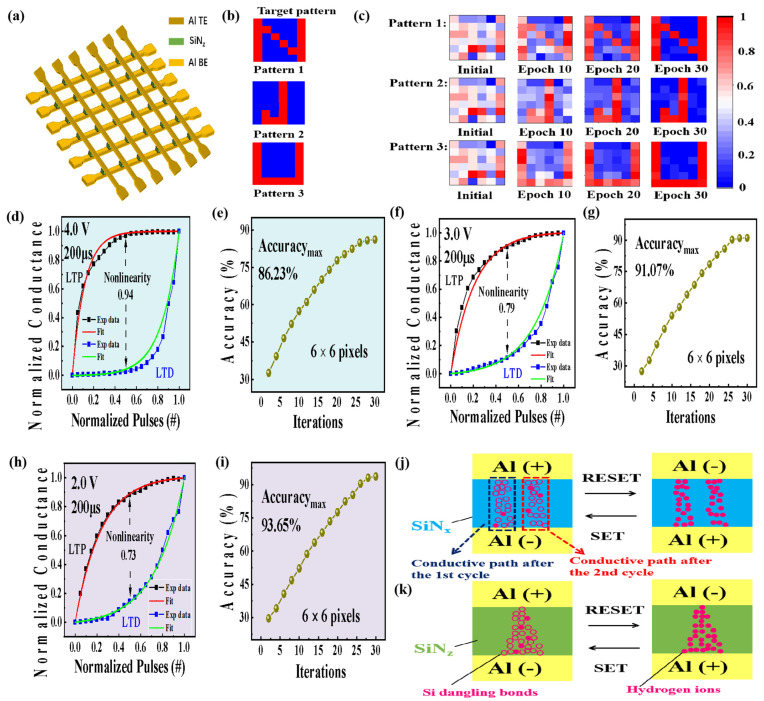
(**a**) A diagram of artificial neuromorphic network based on Al/SiN_z_:H/Al synapse arrays. (**b**) Three visual patterns used for image learning. (**c**) Evolution of the weight maps after 10, 20 and 30 epochs. (**d**,**f**,**h**) Normalized conductance distribution and nonlinearity fitting curves of the Al/SiN_z_:H/Alsynapse under the pulse amplitude of 4.0 V, 3.0 V and 2.0 V, respectively. (**e**,**g**,**i**) The training accuracy of 6 × 6 synapse arrays based on the Al/SiN_z_:H/Al synapse for image learning under the pulse amplitude of 4.0 V, 3.0 V and 2.0 V, respectively. (**j**,**k**) The conductive pathway model of the SiN_x_ synapse with constant Si/N ratio and the SiN_z_ synapse with gradient Si/N ratio. Hydrogen ion and Si dangling bonds are represented by red dot and red circle, respectively.

## Data Availability

All data generated and analyzed during this study are included in this paper. The data presented in this study are available on request from the corresponding author.

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
