# Peer review of "Artificial SiNz:H Synapse Crossbar Arrays with Gradual Conductive Pathway for High-Accuracy Neuromorphic Computing"

_nanomaterials, 2023, doi:10.3390/nano13162362_

Round 1

Reviewer 1 Report

Authors present the fabrication and characterization of SiNz:H resistive switching devices and their properties as synapsys emulators. I find the work interesting, and, from my point of view, it deserves publication in nanomaterials. However, the following points should be corrected, clarified, or commented before publication.

1.       The title, abstract and conclusions induce the reader to think that a hardware neural network has been implemented (and, even, trained). Actually, a neural network is not presented, but a crossbar array of memristors, which are programmed according to three given patterns. So, I suggest changing the title, abstract, …

2.       Authors claim that the conduction mechanism is filamentary, but no evidence is reported. Have been devices with different areas tested?

3.       As authors have fabricated a crossbar array with several devices, it would be very valuable a report about device-to-device variability of the multilevel conductance achieved with the different reset voltages (fig. 2d). Cycle to cycle variability is reported (figure 2f), but not variability in different devices using the same writing schema.

4.       Regarding figures 3b to 3f, authors present them as results linked to short term plasticity (transient enhancement of the weight). However, these same plots could be linked to permanent or non-volatile changes in the device conductance and the reader could not note the difference. If the reached conductance values are volatile, please inform about retention times or provide plots with the read conductance versus time, showing a conductance decay if that is the case.

5.       Regarding figures 3b and 3c, maybe SDDP should be determined (as a function of the pulse width) starting from the same device state. In the current experiment, the comparison between different pulse widths is not “fair” because for the widest pulses, the device has been previously stressed by the previous pulses. Instead, the device should be reset before the application of a new pulse.

6.       Regarding STDP (figure 3h), information about the pulses shape should be provided. In fact, I think that pulse schemas (insets in figure 3h) are confusing: a negative pulse in the post-synapse connection (bottom electrode) is equivalent to a positive pulse in the pre-synapse (top electrode). So, there should not be difference between LTD (depression) and LTP (potentiation).

7.       In figure 4 (and related text), the different programming possibilities (4d to 4i) are presented after the programming results (figure 4c). Please, indicate what programming schema (pulse amplitude) is used for obtaining the results in figure 4c.

8.       What is the training accuracy? Please, define.

Other minor issues:

1.       Please, define DB before its first use (page 2, line 61).

2.       In line 45, all the references 5-18 do not correspond to the mentioned devices (oxide based memristors).

3.       Figure 2f, endurance instead of edurance.

4.       Legend label “Current” in figures 2e-g and 2i is not clear.

5.       Line 179: electric field intensity.

6.       I suggest that authors also explain in the introduction that they have fabricated two kinds of devices (with variable and constant N/si ratio, the SiNz and SiNx devices, respectively).

7.       Figure 3f, in the x-axis legend: delta is upside down.

8.       Accuracy instead of acurracy in plots 4e, 4g and 4i.

9.       Lines 319 and 320: hydrogen instead of oxygen.

 They are listed in my previous comments.

Reviewer 2 Report

This paper mainly presents the fabrication and memristive characteristics of SiNz:H memristor arrays with gradient Si/N ratio. Meanwhile, the authors also performed neural network with crossbar structures and investigated synaptic properties. I think that this paper contains large amount data and valuable information. It can be considered for publication after necessary revisions. The following are some questions that the authors need to clarify.

1. The article presents the results of a study of the endurance (endurance) of memristive structures for only 70 switching cycles. However, this is a modest result compared to metal oxide memristor structures, where the number of resistive switching cycles tends to be in the millions.

2. Experimental dependences of memristive properties from cell to cell, as well as from device to device, are not presented.

3. Since a memristor crossbar was used as an experimental sample, it is not clear how the authors solved the problem of the emergence of uncontrolled current paths through adjacent memristor structures, which can significantly contribute to the results of the measured characteristics.

4. The authors must include the energy consumption during the device's operation.

English presentation in this manuscript need improvement and polishing.

Round 2

Reviewer 2 Report

I am completely satisfied with the responses to comments, I recommend for publication